# Sn-Doped Hematite Films as Photoanodes for Photoelectrochemical Alcohol Oxidation

Vitali A. Grinberg, Victor V. Emets, Alexander D. Modestov *, Aleksey A. Averin, Andrei A. Shiryaev, Inna G. Botryakova and Aleksey V. Shapagin

Frumkin Institute of Physical Chemistry and Electrochemistry, Russian Academy of Sciences, Leninsky Prospekt 31, Building 4, 119071 Moscow, Russia; vitgreen@mail.ru (V.A.G.); victoremets@mail.ru (V.V.E.); alx.av@yandex.ru (A.A.A.); a_shiryaev@mail.ru (A.A.S.); mnemozina86@list.ru (I.G.B.); shapagin@mail.ru (A.V.S.)
* Correspondence: amodestov@mail.ru

**Abstract:** Here, the modification of semiconductor thin film hematite photoanode by doping with Sn ions is reported. Undoped and Sn-doped hematite films are fabricated by the electrochemical deposition of FeOOH from aqueous alkaline electrolyte, followed by calcination in air. The photoanodes were tested in photoelectrocatalytic oxidation of water, methanol, ethylene glycol, and glycerol. It is shown that modification by tin dramatically increased the activity of hematite in the photoelectrochemical oxidation of alcohols upon visible light irradiation. The photoelectrocatalytic activity of Sn-modified hematite increased in the sequence of: $H_2O$ < MeOH < $C_2H_2(OH)_2$ < $C_3H_5(OH)_3$. The quantum yield of photocurrent in the oxidation of alcohols reached 10%. The relatively low photocurrent yield was ascribed to the recombination of photoexcited holes within the hematite layer and on surface states located at the hematite/electrolyte interface. Intensity-modulated photocurrent spectroscopy (IMPS) was used to quantify the recombination losses of holes via surface states. The IMPS results suggested that the hole acceptor in the electrolyte (alcohol) influences photocurrent both by changing the charge transfer rate in the photoelectrooxidation process and by the efficient suppression of the surface recombination of generated holes. Thin-film Sn-modified hematite photoanodes are promising instruments for the photoelectrochemical degradation of organic pollutants.

**Keywords:** Sn-doped hematite films; electrochemical deposition; photoelectrocatalytic oxidation; methanol; ethylene glycol; glycerol



## 1. Introduction

Photoelectrochemical methods are promising, not only for the efficient transformation of solar energy into hydrogen by water splitting [1,2], but also for the degradation of air and water pollutants. Photoelectrocatalytic treatment is applied to numerous substances, which can be divided into two main groups. The first group includes substances harmful to the environment and health, such as dyes, pesticides, pharmaceuticals, herbicides, phenol and its derivatives, etc. Their photoelectrocatalytic degradation aims at converting them to benign compounds. The substances in the second group are oxidized in order to produce higher-value compounds. The photoelectrocatalytic oxidation of alcohols is of particular interest in this connection [3–6].

A broad variety of semiconductor materials based on n- and p-type oxides is used as the main part of photoelectrochemical devices, namely $TiO_2$ [7], $WO_3$ [8,9], $\alpha$-$Fe_2O_3$ [10–12], ZnO [13,14], $BiVO_4$ [15,16], $Cu_2O$ [17], $CuRhO_2$ [18]. Photoanodes based on titanium dioxide and zinc oxide are of considerable interest because of their high corrosion resistance, low cost, and nontoxicity [6,19]. However, both of these semiconductors are characterized by wide band gaps (3.2 and 3.37 eV, respectively), and thus absorb only a minor fraction (~4%) of solar irradiation reaching the Earth's surface. Therefore, the development of photoelectrocatalytic systems absorbing a larger part of the visible solar spectrum is an important challenge. Hematite, $\alpha$-$Fe_2O_3$, is an n-type semiconductor with a much smaller

band gap of $E_g$ = 2.2 eV. It absorbs light with wavelengths shorter than 600 nm and may be a promising photoanode material for some photoelectrochemical oxidation reactions.

Hematite is stable in aqueous solutions at pH > 3 [1,20]. Hematite photoanodes have been used for the photoelectrochemical decomposition of water and degradation of organic pollutants [21,22] and glycerol [3,23]. In these processes, a high level of recombination losses of photoexcited holes, reducing the efficiency of light utilization, has been observed. The losses are caused by several processes, including the short diffusion length of the photogenerated holes, low rate of charge transfer to the species adsorbed on the semiconductor surface, and low conductivity [24,25]. However, the losses can be significantly reduced by the modification of hematite's structure and doping. Previously, we reported an effective reduction in losses from a thin-film hematite photoanode by doping with titanium or by modification with zinc oxide. These photoanodes were effective in the photoelectrocatalytic degradation of methanol, ethylene glycol, and glycerol under visible light irradiation [26–28]. Hematite doping by Zr, Cr, Mo, Zn, Cd, Ni, Pt, Ti, Ge, Si, and Sn has been proposed [28–31]. Sn-doped hematite has been widely studied due to the relatively high chemical stability of Sn precursors, such as $SnCl_2$ and $SnCl_4$, in aqueous solutions and simplicity of the doping process. Various methods have been used to produce doped hematite films, including electrochemical deposition, spray pyrolysis, the sol-gel method, co-evaporation of iron and tin in an oxygen atmosphere, hydrothermal and high-temperature annealing, and pulsed laser deposition (PLD) [32–37]. Electrochemical deposition methods are advantageous as they do not employ expensive equipment and harsh conditions for film growth [38–42]. During high-temperature annealing, the film becomes crystalline and $Sn^{4+}$ ions diffuse into the hematite lattice by replacing iron ions [43,44]. The charge balance can be maintained by the formation of cation vacancies or by the partial reduction of $Fe^{3+}$ to $Fe^{2+}$. The concentration of the Sn dopant in the films can be controlled by selecting the chemical composition of the initial electrolytes.

A current density of 2.8 mA cm$^{-2}$ and light conversion efficiency of 0.24% were reached in photoelectrochemical water decomposition on a Sn-doped hematite film at potential $E$ = 1.24 V versus reversible hydrogen potential (vs. RHE) [42]. The record high photocurrent for water oxidation on doped hematite is 5.7 mA cm$^{-2}$ at $E$ = 1.23 V (vs. RHE) and standard sunlight intensity (1 Sun = 100 mW cm$^{-2}$). However, this value is still well below the theoretical maximum [45]. The photoelectrochemical decomposition of water on tin-doped film hematite photoanodes is well-studied [46–50]. However, only a few studies have been devoted to the photoelectrocatalytic degradation of organic compounds [3,21–23]. The photoelectrochemical oxidation of alcohols may proceed via two different paths. In the first case, the interaction of a photoexcited hole with an adsorbed organic molecule is the primary oxidation reaction. In the second case, the primary reaction is the photoelectrochemical oxidation of water with the formation of a highly active hydroxyl radical. The alcohol oxidation product is formed by the reaction of the radical with an alcohol molecule in the vicinity of the electrode. The actual reaction path of alcohol oxidation depends on processes within the photoanode material and on the interactions of organic species with the photoanode surface. Alcohol molecules with different structures occupy different numbers of adsorption sites [26–28]. The efficiency of photocurrent generation is largely determined by the recombination of photoexcited holes within the semiconductor and on its surface via surface states.

In the present work, we investigated the activity of Sn-modified hematite films in the reaction of photoelectrocatalytic degradation of methanol, ethylene glycol, and glycerol under visible light illumination. Photoanodes were fabricated by the electrochemical deposition of the oxide films on a glass substrate coated with a conductive layer of fluorine-stabilized tin dioxide (FTO glass). The deposition of hematite was accompanied by the incorporation of different amounts of $Sn^{4+}$ into the film. The influence of the nature of alcohol, as an acceptor of photoexcited holes, on the charge carrier recombination rate in these films is elucidated. Note that the current study does not address the composition of the products of photoelectrooxidation of the mentioned alcohols.

## 2. Results and Discussion

The photoanodes were fabricated by electrodeposition at a controlled potential $E = -0.35$ V vs. Ag/AgCl reference for 150 s unless otherwise stated. The samples with pure hematite films were designated as ($\alpha$-Fe$_2$O$_3$). Photoanodes with films of Sn$^{4+}$-modified $\alpha$-Fe$_2$O$_3$ were obtained at the same potential $E = -0.35$ V vs. Ag/AgCl (deposition time of 150 s) in the presence of various amounts of SnCl$_4$ in the electrodeposition electrolyte (see Section 3). Owing to the presence of hydrogen peroxide in the electrolyte of iron oxyhydroxide FeOOH electrodeposition, the divalent tin in the electrolyte is transformed into the tetravalent state (see Supplementary Materials Figures S1–S3). The photoanodes were designated as ($\alpha$-Fe$_2$O$_3$ + 5% Sn$^{4+}$), ($\alpha$-Fe$_2$O$_3$ + 10% Sn$^{4+}$), and ($\alpha$-Fe$_2$O$_3$ + 20% Sn$^{4+}$). The indicated Sn$^{+4}$ content represents the SnCl$_2$/FeCl$_3$ ratio in the film-forming electrolyte. All samples were annealed in air at 500 °C for 2 h and at 750 °C for 10 min.

### 2.1. X-ray Diffraction Patterns

Figure 1 shows the X-ray diffraction patterns of the conductive FTO glass substrate and of the hematite films coated on FTO glass.

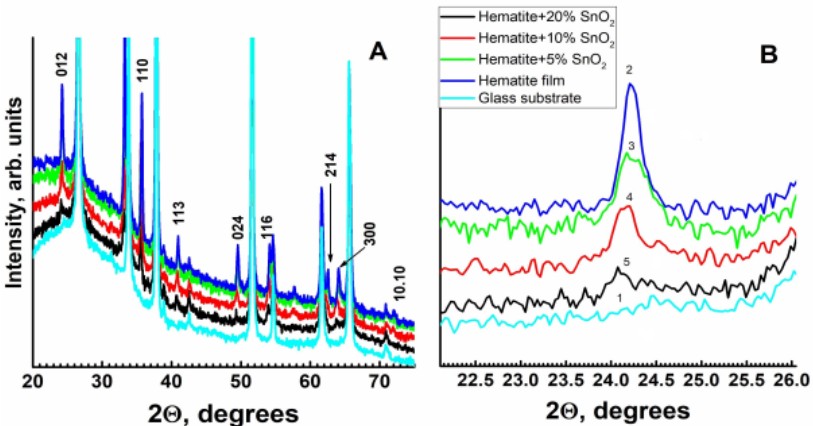

**Figure 1.** X-ray diffraction patterns of the studied samples: (1)—FTO; (2)—($\alpha$-Fe$_2$O$_3$); (3)—($\alpha$-Fe$_2$O$_3$ + 5% Sn$^{4+}$); (4)—($\alpha$-Fe$_2$O$_3$ + 10% Sn$^{4+}$); (5)—($\alpha$-Fe$_2$O$_3$ + 20% Sn$^{4+}$). (**A**)—Overall view, Miller indices of principal hematite reflections are indicated. (**B**)—Zoomed-in region of hematite 012 reflection showing the progressive shift of the peak and its intensity decrease with the increase in the Sn$^{4+}$ doping level.

The phase composition of the films was assessed by X-ray diffraction in Bragg–Brentano (reflection) geometry. The pattern of the FTO glass slide was dominated by SnO$_2$ (curve (1) in Figure 1). The diffraction patterns of the samples confirm the formation of hematite film (see Figure 1). A detailed examination of the films prepared with a gradually increasing amount of tin in the reaction media shows that Sn addition led to progressive moderate swelling of the hematite lattice (unit cell volume increased by ~0.5% at the highest Sn loading). This behavior is consistent with the assumption of the formation of Sn solid solution in hematite. (Direct evidence of Sn incorporation into the film is provided in the EDX and XPS studies in the Supplementary Materials). The relative intensity of the hematite reflections decreases with the Sn loading largely due to the smaller thickness of the films. In addition, the formation of an X-ray amorphous Fe-containing phase along with hematite cannot be excluded. According to the Fe$_2$O$_3$-SnO$_2$ equilibrium phase diagram [51], at temperatures below 1000 °C, the solubility of SnO$_2$ in hematite does not exceed a few percent; both phases—$\alpha$-Fe$_2$O$_3$ and SnO$_2$—coexist at 475 °C [52]. Studies on SnO$_2$ solubility in hematite thin films (e.g., [37] and refs. therein) are consistent with the equilibrium phase diagram.

## 2.2. Raman Spectra

Figure 2 shows the Raman spectra of the deposited films. The observed characteristic peaks at 222, 292, 409, 494, and 608 cm$^{-1}$ are close to the exact characteristic phonon vibration modes of $\alpha$-Fe$_2$O$_3$ [53]; small shifts may arise from substrate-induced strain. A characteristic shoulder at 658 cm$^{-1}$ was observed for all samples; its intensity increased with the increase in Sn$^{4+}$ doping. While this peak might correspond to magnetite (Fe$_3$O$_4$) admixture (e.g., [54]), it is more likely that it is related to the disorder-activated IR-active mode of hematite [55,56]. The intensity variations of this shoulder with the concentration of added Sn$^{4+}$ can be related to the disorder of hematite lattice induced by the dopant.

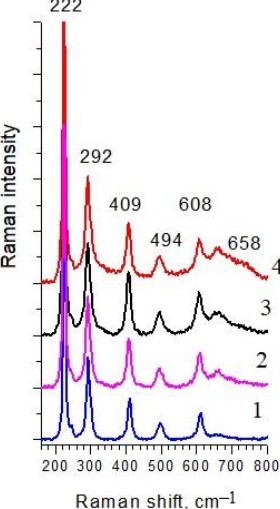

**Figure 2.** Raman spectra of the film samples: (1)—($\alpha$-Fe$_2$O$_3$); (2)—($\alpha$-Fe$_2$O$_3$ + 5% Sn$^{4+}$); (3)—($\alpha$-Fe$_2$O$_3$ + 10% Sn$^{4+}$); (4)—($\alpha$-Fe$_2$O$_3$ + 20% Sn$^{4+}$).

## 2.3. Absorption Spectra

The absorption ($\alpha$) spectra of the original and Sn-modified hematite films in the visible spectral range are shown in Figure 3. To eliminate the effect of the film thickness, the curves shown in Figure 3a were normalized to [0, 1] (Figure S6). The band gap of the electrodeposited films was estimated using Tauc coordinates [43,57] constructed for normalized curves (see Figure 3b). The direct band gap of a semiconductor ($E_g$) can be obtained by the extrapolation of the linear part of the function ($\alpha h\nu)^2$ to the X axis (photon energy, $h\nu$). As shown in Figure 3b, the modification of $\alpha$-Fe$_2$O$_3$ with Sn$^{4+}$ virtually did not change the band gap energy; the $Eg$ values for the studied samples were close to 2.19 eV.

With the increase in tin concentration, the light absorption decreased in comparison with pure hematite, which is consistent with the results obtained in [37]. The greatest effect was observed for the sample ($\alpha$-Fe$_2$O$_3$ + 20% Sn$^{4+}$/FTO), which was partly explained by the smallest thickness of the film. The $E_g$ value obtained for the Sn-modified hematite samples was very close to the value of the band gap $E_g$ = 2.14 eV for the $\alpha$-Fe$_2$O$_3$ calcined at a high temperature [36].

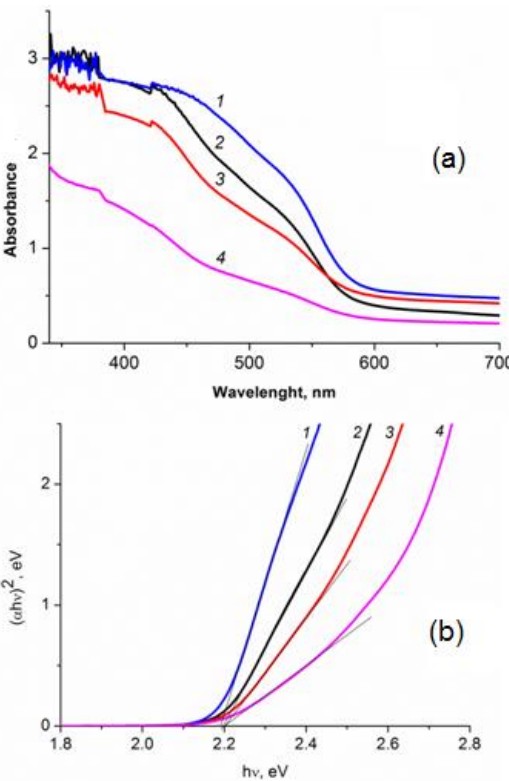

**Figure 3.** Absorption spectra (**a**) and dependence of $(\alpha h\nu)^2$ on photon energy $h\nu$ (**b**) for the hematite film photoanodes without $Sn^{4+}$ (1)—($\alpha$-$Fe_2O_3$); and with different amount of $Sn^{4+}$ (2)—($\alpha$-$Fe_2O_3$ + 5% $Sn^{4+}$); (3)—($\alpha$-$Fe_2O_3$ + 10% $Sn^{4+}$), and (4)—($\alpha$-$Fe_2O_3$ + 20% $Sn^{4+}$).

### 2.4. Morphology and Thickness of the Electrodeposited Films

The thickness of the electrodeposited films was determined on a thin films measurement system MProbe 20 (Semiconsoft Inc., Southborough, MA, USA). The film thickness decreased with the increase in the $Sn^{+4}$ concentration in the electrolyte. The thicknesses of the ($\alpha$-$Fe_2O_3$), ($\alpha$-$Fe_2O_3$ + 5% $Sn^{4+}$), ($\alpha$-$Fe_2O_3$ + 10% $Sn^{4+}$), and ($\alpha$-$Fe_2O_3$ + 20% $Sn^{4+}$) films were ~300 nm, ~250 nm, ~200 nm, and ~150 nm, respectively.

The morphology of the deposited films was studied by scanning electron microscopy (SEM) with a JEOL JSM-6060 SEM (JEOL, Tokyo, Japan). Representative SE (Secondary Electrons) images are shown in Figure 4. According to the SEM images, the films consisted of crystallites with an average diameter ranging from 80 nm to 300 nm. Prior to heat treatment, the surface was quite smooth and yellow in color, i.e., iron oxyhydroxide was evenly deposited on the surface of the conductive glass. Cracks were formed on the surface of the hematite film as a result of stress that arose in heat treatment at 500 °C (2 h) and 750 °C (10 min). With an increase in the concentration of tin in the electrolyte to 20%, the thickness of the film decreased and the network of cracks practically disappeared. The large white "particles" in Figure 4 are clusters of Fe oxide nanoparticles. Their exact composition is difficult to determine because of their rather small size. The high brightness in the SE mode is due to the pronounced topography and well-developed surface, providing numerous spots with enhanced emission of secondary electrons. The same "particles" are also visible on AFM scans (see Figure 5).

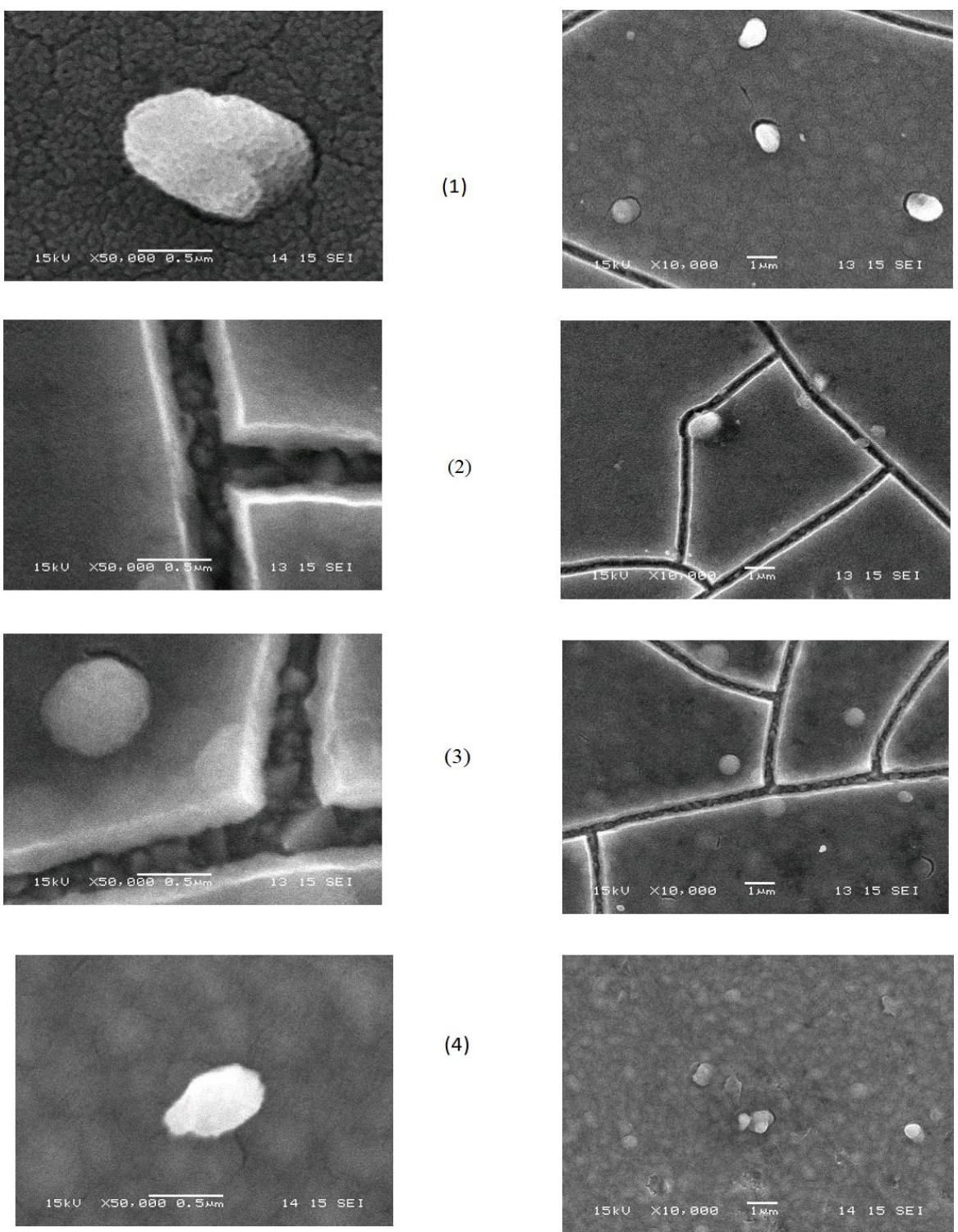

**Figure 4.** SEM images of hematite films prepared by electrodeposition at $E = -0.35$ V vs. Ag/AgCl: (**1**)—($\alpha$-Fe$_2$O$_3$); (**2**)—($\alpha$-Fe$_2$O$_3$ + 5% Sn$^{4+}$); (**3**)—($\alpha$-Fe$_2$O$_3$ + 10% Sn$^{4+}$); (**4**)—($\alpha$-Fe$_2$O$_3$ + 20% Sn$^{4+}$). All samples were annealed in air at 500 $^\circ$C for 2 h and 750 $^\circ$C for 10 min.

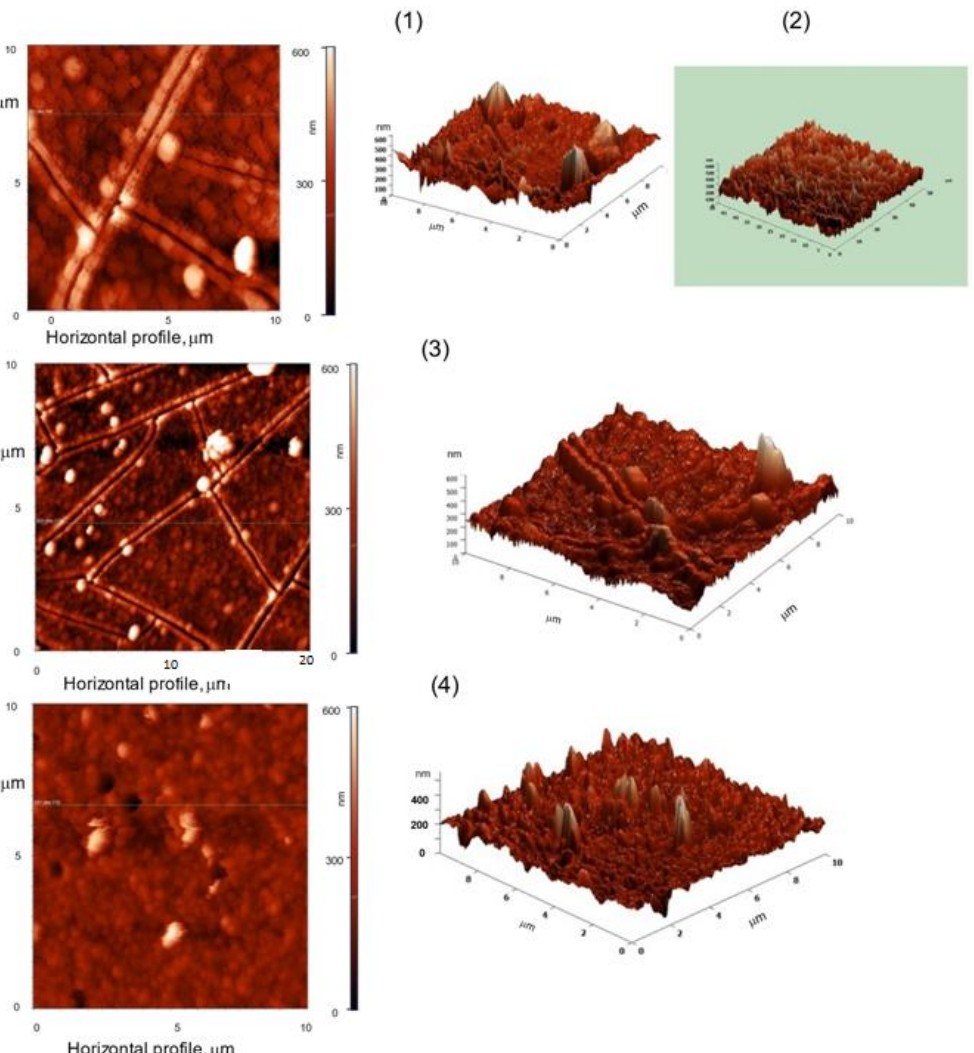

**Figure 5.** AFM topography of hematite film (**1**)—$\alpha$-Fe$_2$O$_3$ and films with different amounts of Sn$^{4+}$; (**2**)—($\alpha$-Fe$_2$O$_3$ + 5% Sn$^{4+}$); (**3**)—($\alpha$-Fe$_2$O$_3$ + 10% Sn$^{4+}$); (**4**)—($\alpha$-Fe$_2$O$_3$ + 20% Sn$^{4+}$) horizontal profile and 3D view. The 3D scan areas are: $10 \times 10$ $\mu$m$^2$ for (**1**), (**3**), and (**4**), and $50 \times 50$ $\mu$m$^2$ for (**2**).

### 2.5. Atomic Force Microscopy (AFM)

The morphology of the doped hematite samples was also studied using AFM. Figure 5 shows images of the films formed with different concentrations of Sn$^{4+}$ in the electrolyte: (1)—($\alpha$-Fe$_2$O$_3$), (2)—($\alpha$-Fe$_2$O$_3$ + 5% Sn$^{4+}$), (3)—($\alpha$-Fe$_2$O$_3$ + 10% Sn$^{4+}$), and (4)—($\alpha$-Fe$_2$O$_3$ + 20% Sn$^{4+}$).

The AFM images correlate well with the SEM results. The grain sizes of the hematite and tin-doped hematite samples varied within 100–300 nm (Figure 4). The images clearly show cracks and surface defects. On the ($\alpha$-Fe$_2$O$_3$ + 20% Sn$^{4+}$) film, the cracks were practically not visible, apparently because the thin film was less prone to cracking under high-temperature treatment.

### 2.6. Photoelectrochemical Oxidation of Water, Methanol, Ethylene Glycol, and Glycerol on Sn Modified Hematite Photoanodes

The preliminary photoelectrochemical experiments in the supporting 0.1 M KOH electrolyte (Figure 6) showed that the doping of hematite films with tin resulted in an increase in water oxidation photocurrents (at $E$ = 0.5 V vs. Ag/AgCl) by a factor of 3.5–9 compared with the $\alpha$-Fe$_2$O$_3$ photoanode. Thus, doping with tin increased the efficiency of water photoelectrolysis on hematite.

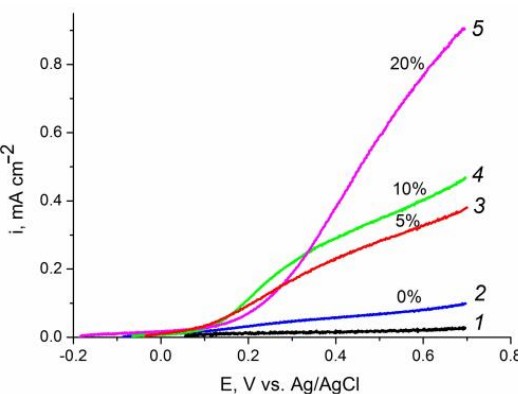

**Figure 6.** Voltammetry curves measured in 0.1 M KOH at photoanodes coated with hematite films: (1)—background (dark) curve; (2–5)—curves for the samples with various amounts of added $Sn^{4+}$ at 0, 5, 10, and 20%, respectively. Curves (2–4) were measured under photoanode irradiation with visible light of 100 mW cm$^{-2}$ power. The potential scan rate is 10 mV s$^{-1}$. The dark curves for all four samples practically coincide with curve (1).

Figure 7 shows the voltammetry curves of the photoelectrooxidation of methanol, ethylene glycol, and glycerol in 0.1 M KOH at the ($\alpha$-Fe$_2$O$_3$ + 10% Sn$^{4+}$) photoanode. Well-pronounced waves of direct photoelectrochemical oxidation were observed for all studied alcohols. The curves shifted to negative potential values compared with the curves shown in Figure 6. In comparison with the curves measured in 0.1 M KOH, the photocurrents at $E = 1.23$ V vs. RHE ($E = 0.27$ V vs. Ag/AgCl) in electrolytes containing 20% methanol, ethylene glycol, and glycerol were higher by factors of 3, 5, and 5.5, respectively. The observed photocurrent increase implies an increase in the photoelectrooxidation reaction rates in the order of: $H_2O < CH_3OH < C_2H_2(OH)_2 < C_3H_5(OH)_3$. The photoelectrooxidation rates of these alcohols on ($\alpha$-Fe$_2$O$_3$) photoanode changed in the same sequence (Figure S5). A comparison of the data provided in Figure 7 and Figure S7 shows that, in the studied alcohol solutions, the photocurrents at ($\alpha$-Fe$_2$O$_3$ + 10% Sn$^{4+}$) photoanode were higher than those at ($\alpha$-Fe$_2$O$_3$) by an order of magnitude at $E = 1.23$ V vs. RHE ($E = 0.27$ V vs. Ag/AgCl). Taking into account similar values of the band gap for ($\alpha$-Fe$_2$O$_3$) and ($\alpha$-Fe$_2$O$_3$ + 10% Sn$^{4+}$) (Figure 3a,b), this result can be explained by an increase in the density of surface states (SS) on the hematite photoelectrode by Sn$^{4+}$ modification [58]. An increase in the SS density decreased the recombination of photogenerated charges (electrons and holes) and increased the efficiency of charge transfer to water and alcohol molecules.

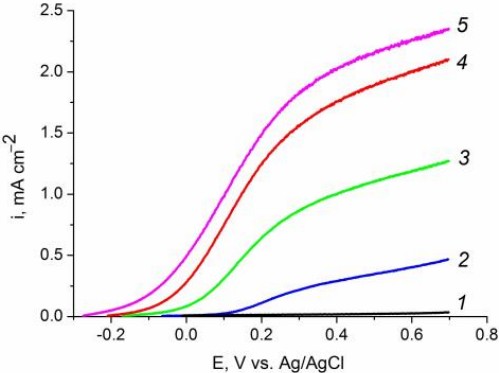

**Figure 7.** Voltammograms of ($\alpha$-Fe$_2$O$_3$ + 10% Sn$^{4+}$) film photoanode. (1)—Dark voltammetry in 0.1 M KOH: (2–5) measured under visible light irradiation with a power density of 100 mW cm$^{-2}$ in aqueous 0.1 M KOH; (2)—in the absence of alcohol. Curves (3–5) were measured in the presence of 20% alcohol: (3)—CH$_3$OH; (4)—C$_2$H$_4$(OH)$_2$; (5)—C$_3$H$_5$(OH)$_3$. The potential scan rate was 10 mV s$^{-1}$. Voltammograms under dark conditions for the alcohol-containing solutions virtually coincided with (1).

The current–voltage characteristics of the hematite films modified with 5 and 10% of $Sn^{4+}$ in reactions of the photoelectrocatalytic oxidation of methanol, ethylene glycol, and glycerol are shown in Figures S8 and S9, respectively. In the presence of alcohols in the electrolyte, two processes contributed to the photocurrent value—the photooxidation of water and of the alcohols. To find the amount of tin additive that provided the most efficient photoelectrocatalytic oxidation of the studied alcohols, the corresponding partial current–voltage curves were calculated by subtracting the water oxidation photocurrent from the actually measured photocurrent values at all potentials. The calculated partial alcohol oxidation curves at the studied photoanodes are shown in Figure 8. It can be seen that the maximum partial photocurrent densities for each of the alcohols were observed at the ($\alpha$-$Fe_2O_3$ + 10% $Sn^{4+}$) photoanode. At the ($\alpha$-$Fe_2O_3$ + 5% $Sn^{4+}$) photoanode, a slight decrease in the partial photocurrent was observed; at the photoanode with 20% $Sn^{4+}$, the effect was more pronounced. The latter case can be explained by the high efficiency of the competition process in the photoelectrooxidation of water (Figure 6) and by the decreases in the film thickness and light absorption (Figure 3). The presence of an amorphous phase in the ($\alpha$-$Fe_2O_3$ + 20% $Sn^{4+}$) sample and non-optimal number of surface states also negatively influenced the partial photocurrent values. The obtained data show that the addition of 10% $Sn^{4+}$ to the hematite photoanode ensured the maximum efficiency of the photoelectrocatalytic oxidation of $CH_3OH$, $C_2H_2(OH)_2$, and $C_3H_5(OH)_3$.

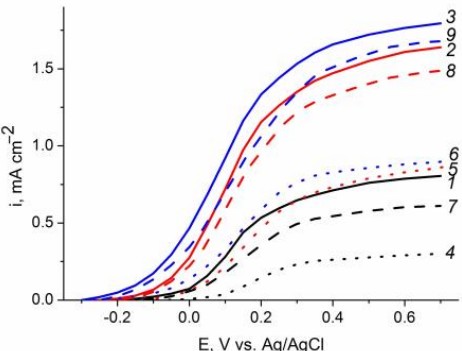

**Figure 8.** Partial voltammetry curves of the photoanodes in aqueous 0.1 M KOH. (1–3): ($\alpha$-$Fe_2O_3$ + 10% $Sn^{4+}$); (4–6): ($\alpha$-$Fe_2O_3$ + 5% $Sn^{4+}$); (7–9): ($\alpha$-$Fe_2O_3$ + 20% $Sn^{4+}$). Twenty percent of the following alcohols was added to the solutions: $CH_3OH$—(1), (4), and (7); $C_2H_4(OH)_2$—(2), (5), and (8); $C_3H_5(OH)_3$—(3), (6), and (9). The measurements were conducted under visible light irradiation of 100 mW $cm^{-2}$ power density. The potential scan rate was 10 mV $s^{-1}$.

Figure 9 shows the corresponding partial voltammograms for two ($\alpha$-$Fe_2O_3$ + 10% $Sn^{4+}$) photoanodes with different thicknesses. The photoanodes were formed by electrodeposition for 150 s (A) and for 75 s (B); their thicknesses differed by a factor of ~2. The current–voltage characteristics of photoanode (B) in the reactions of photoelectrocatalytic oxidation of methanol, ethylene glycol, and glycerol are shown in Figure S10. However, the film thickness had little effect on the partial voltammetry of the photooxidation of alcohols. The recombination losses seemed to increase in the thicker film due to the short diffusion length of the photogenerated holes.

The stability of the photoanode ($\alpha$-$Fe_2O_3$ + 10% $Sn^{4+}$) with the highest photoelectrocatalytic properties was studied in the oxidation of glycerol (Figure S11). The photocurrent of glycerol oxidation at potentials of 0.4 and 0.6 (vs. Ag/AgCl) upon irradiation by a sunlight simulator of 1 Sun virtually did not change in the course of 30 min measurement.

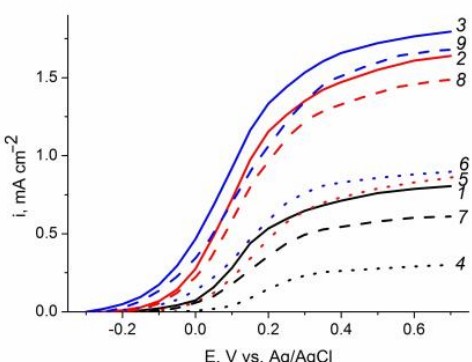

**Figure 9.** Partial voltammetry curves measured at photoanodes ($\alpha$-Fe$_2$O$_3$ + 10% Sn$^{4+}$) of different hematite thickness (A)—(1–3) and (B)—(4–6) in 0.1 M KOH in the presence of 20% alcohol: CH$_3$OH—(1), (4), and (7); C$_2$H$_4$(OH)$_2$—(2), (5), and (8); C$_3$H$_5$(OH)$_3$—(3), (6), and (9). The measurements were conducted under visible light irradiation of 100 mW cm$^{-2}$ power density. The potential scan rate was 10 mV s$^{-1}$.

Common to all studied photoanodes was an increase in the photoelectrooxidation current in the sequence of: H$_2$O < CH$_3$OH < C$_2$H$_4$(OH)$_2$ < C$_3$H$_5$(OH)$_3$. This can be explained by the influence of the chemical nature of the organic species adsorbed on the surfaces of photoanodes, both on the rate constant of photooxidation and on the recombination of photoexcited charge carriers at the surface. The recombination of photoexcited charge carriers was evidenced by the shape of the photocurrent transients shown in Figure 10. In 0.1 M KOH, switching the light on and off resulted in a sharp jump in the photocurrent, which quickly dropped to a stationary value (Figure 10, curve 1) due to the partial recombination of photogenerated holes. The indicated current jump and, consequently, the associated recombination losses decreased significantly upon the addition of 20% CH$_3$OH to the electrolyte (Figure 10, curve 2) and practically disappeared if 20% C$_2$H$_4$(OH)$_2$ or 20% C$_3$H$_5$(OH)$_3$ are added (Figure 10, curves (3) and (4), respectively). The surface states play a much more important role than the bulk recombination.

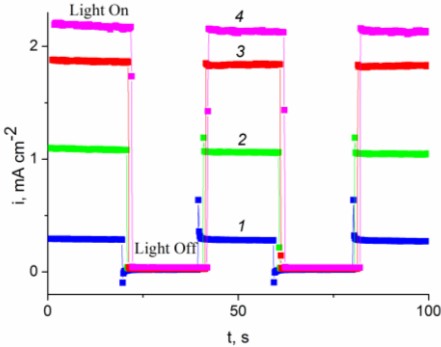

**Figure 10.** Photocurrent transients measured at $E$ = 0.5 V (vs. Ag/AgCl) on the ($\alpha$-Fe$_2$O$_3$ + 10% Sn$^{4+}$) photoanode under periodic irradiation by visible light of 100 mW cm$^{-2}$ power in 0.1 M KOH (1) and in the presence of 20% CH$_3$OH (2); 20% C$_2$H$_4$(OH)$_2$ (3); 20% C$_3$H$_5$(OH)$_3$ (4).

For the ($\alpha$-Fe$_2$O$_3$ + 10% Sn$^{4+}$) photoanode, the observed partial photocurrents of glycerol oxidation at $E$ = 0.27 V vs. Ag/AgCl (1.23 V vs. RHE) in 0.1 M KOH was at least four times higher than those reported for cobalt-promoted zinc oxide and hematite photoanodes (1.5 mA cm$^{-2}$ vs. 0.4 mA cm$^{-2}$ [6] and 0.25 mA cm$^{-2}$ [23], respectively). The partial currents of glycerol photoelectrooxidation were 1.5 times higher than the currents obtained with a BiVO$_4$ photoanode, on which a current density of glycerol oxidation of up to 1.0 mA cm$^{-2}$ was obtained in a 0.5 M Na$_2$SO$_4$ solution at pH 5 and pH 7 [3]. Note that the partial currents of methanol photooxidation at $E$ = 0.27 V vs. Ag/AgCl in 0.1 M KOH on

a tin-promoted hematite photoanode were an order of magnitude higher than the currents of methanol oxidation obtained on a hematite photoanode in an aqueous solution [5].

### 2.7. Evaluation of Recombination Losses in the Photoelectrooxidation of Alcohols

The influence of the nature of the alcohol on recombination losses was studied in detail, with the ($\alpha$-Fe$_2$O$_3$ + 10% Sn$^{4+}$) photoanode manifesting the highest activity (Figures 7 and 8). Figure 11 shows the wavelength dependence of the quantum efficiency (IPCE) of current generation at the photoanode in 0.1 M KOH + 20% glycerol. The photocurrent was generated by visible light with 350–580 nm wavelength; above 580 nm the photoactivity is negligible. The IPCE spectrum agreed well with the light absorption spectrum of this sample (Figure 3a, curve 3).

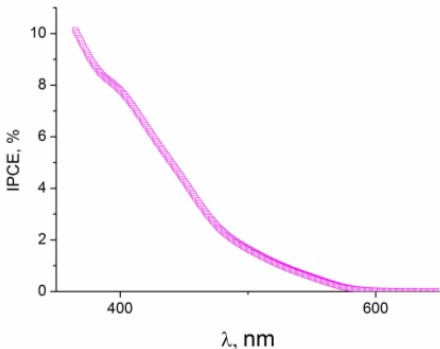

**Figure 11.** IPCE spectrum of photocurrent at ($\alpha$-Fe$_2$O$_3$ + 10% Sn$^{4+}$) photoanode at $E$ = 0.5 V (vs. Ag/AgCl) in aqueous 0.1 M KOH + 20% C$_3$H$_5$(OH)$_3$.

Intensity-modulated photocurrent spectroscopy (IMPS) [59–61] was used to quantify the recombination losses of photogenerated holes in the hematite layer in the photooxidation of water and alcohols. Monochromatic 452 nm irradiation was used, as it produces a rather high value of IPCE (Figure 11) and ensures good measurement accuracy. The IMPS spectra were measured in 0.1 M KOH solution (Figure S12) and in 0.1 M KOH solution containing 20% methanol, ethylene glycol, or glycerol. The high-frequency intersection of the IMPS curve with the X axis provides the value of the total photogenerated current $I_2$. The low-frequency intercept of IMPS curve provides the $I_1$ value, which corresponds to the oxidation of the species present in the electrolyte. Owing to recombination losses, I$_1$ < I$_2$.

To quantify the recombination losses, the IMPS curves were normalized to $I_2$ (Figure 12). In 0.1 M KOH at $E$ = 0.5 V (vs. Ag/AgCl), the photocurrent of water oxidation reached 46% of the total photocurrent, $I_1/I_2$ = 0.46 (curve 1); the recombination losses (($I_2 - I_1$)/$I_2$ = 0.54) reached 54%. As follows from the curve (3) in Figure 12, the addition of 20% CH$_3$OH to 0.1 M KOH reduced the recombination losses to 11%, while the addition of 20% C$_2$H$_4$(OH)$_2$ or 20% C$_3$H$_5$(OH)$_3$ to 0.1 M KOH virtually eliminated the losses (curves 3 and 4, respectively).

The IMPS measurements enabled the calculation of the recombination rate constants $K_{rec}$ and charge transfer rate constants $K_{ct}$. The low-frequency intercept of the IMPS spectrum (the ratio $I_1/I_2$ in Figure 12) was related to these constants, as $I_1/I_2 = K_{ct}/(K_{rec} + K_{ct})$. The light intensity modulation frequency corresponding to the maximum of the semicircle located in the first quadrant ($f_{max}$ in Figure 12) made it possible to find the sum ($K_{rec} + K_{ct}$) from the equation: $2\pi f_{max} = (K_{rec} + K_{ct})$.

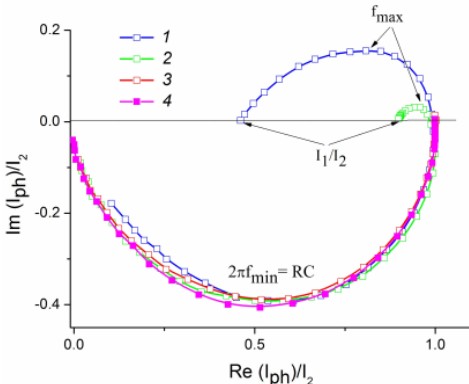

**Figure 12.** Normalized IMPS dependences measured on the ($\alpha$-Fe$_2$O$_3$ + 10% Sn$^{4+}$)/FTO photoanode irradiated by monochromatic 452 nm light with 14 mW cm$^{-2}$ power density. The measurements performed at $E$ = of 0.5 V (vs. Ag/AgCl) were in 0.1 M KOH—(1); and in the presence of 20% alcohols in 0.1 M KOH: CH$_3$OH—(2); C$_2$H$_4$(OH)$_2$—(3); C$_3$H$_5$(OH)$_3$—(4).

The high-frequency part of the IMPS curve was marred by the influence of the $RC$ time constant of the electrochemical cell (Figure 12), which arose from the series resistance of the FTO glass ($R$) and the space charge capacitance of the oxide film ($C$). The $f_{min}/f_{max}$ ratio for the ($\alpha$-Fe$_2$O$_3$ + 10% Sn$^{4+}$) photoanode in the studied electrolytes was ~50. The rather high value of the $f_{min}/f_{max}$ ratio provided sufficient accuracy of the calculated values of $K_{rec}$ and $K_{ct}$ [46]. The $K_{rec}$ and $K_{ct}$ values calculated from the IMPS data for photoelectrooxidation at the ($\alpha$-Fe$_2$O$_3$ + 10% Sn$^{4+}$) photoanode were 10.8 and 9.7 s$^{-1}$ for water, respectively. For methanol electrooxidation, the $K_{rec}$ and $K_c$ values were 2.2 and 17.7 s$^{-1}$, respectively. For ethylene glycol and glycerin, as follows from Figure 12, $K_{ct} \gg K_{rec}$. Therefore, there was a significant increase in $K_{ct}$ in the sequence of: H$_2$O < MeOH < C$_2$H$_2$(OH)$_2$ < C$_3$H$_5$(OH)$_3$. In turn, the accelerated consumption of holes from the surface states of the photoanode in the photoelectrooxidation of alcohols was the reason for the decrease in $K_{rec}$. It can be assumed that the decrease in recombination losses in the photoelectrooxidation reaction in the sequence of alcohols of MeOH < C$_2$H$_2$(OH)$_2$ < C$_3$H$_5$(OH)$_3$ was caused by an increase in their adsorption on the photoanode surface.

### 3. Experimental Section

#### 3.1. Materials

Chemically pure (>99%) ferric chloride (FeCl$_3$*6H$_2$O), potassium fluoride (KF*2H$_2$O), potassium chloride (KCl), tin chloride SnCl$_2$*2H$_2$O, and 35% hydrogen peroxide (H$_2$O$_2$) from Sigma-Aldrich (Burlington, MA, USA) were used in the film coating without further purification. To manufacture the photoanodes, glasses with electrically conductive fluorine-stabilized tin dioxide coatings (specific resistance $\approx$ 7 $\Omega$ cm$^{-2}$) (F:SnO$_2$, FTO) (Sigma-Aldrich) were used.

#### 3.2. Preparation of Hematite Films

Photoanodes were produced by the deposition of polycrystalline $\alpha$-Fe$_2$O$_3$ films on FTO-coated glass substrates. Prior to deposition, the substrates were cleaned by ultrasonication in acetone, isopropanol, and distilled water baths (15 min in each). The clean FTO glass was fixed in a Teflon frame of the electrochemical cell so that the surface exposed to the electrolyte was 1 cm$^2$. Pt–Ir alloy plate (Ir 10%) of 8 cm$^2$ area was used as a counter electrode. A silver plate anodized in chloride-containing electrolyte served as the Ag/AgCl reference electrode. The electrodes were fixed to the glass cover of a 100 mL electrochemical cell. The cell temperature was controlled by circulating water between the thermostat and water jacket of the cell. The cell was filled with an electrolyte solution (12.5 mM FeCl$_3$, 50 mM KF, 0.1 M KCl, and 1 M H$_2$O$_2$) just before $\alpha$-Fe$_2$O$_3$ deposition. The hematite film was electrodeposited for 150 or 75 s at $E$ = –0.35 V (vs. Ag/AgCl) at 70 °C. The electric

charge that passed during 150 s of oxide film deposition was 1.4–1.6 C cm$^{-2}$. The reactions, which occurred during electrodeposition, are described in refs. [39,40]:

$$F^- + Fe^{3+} \rightarrow FeF^{2+} \tag{1}$$

$$H_2O_2 + 2e^- \rightarrow 2OH^- \tag{2}$$

$$FeF^{2+} + 3OH^- \rightarrow FeOOH + F^- + H_2O. \tag{3}$$

The overall reaction can be represented by the equation:

$$FeF^{2+} + 3OH^- \rightarrow FeOOH + F^- + H_2O. \tag{4}$$

As a result of the electrodeposition, a yellow FeOOH film was produced on the surface of the FTO slide. It was thoroughly washed with distilled water, dried at room temperature, and then annealed in air in a tube furnace at 500 °C for 2 h. Then, the temperature in the furnace was raised to 750 °C for additional treatment for 10 min. After cooling in the furnace, samples with a 300–150 nm thick uniform red $\alpha$-Fe$_2$O$_3$ film, depending on the electrolyte composition and electrolysis duration, were obtained.

### 3.3. Modification of Hematite Films

To prepare photoanodes with Sn-modified hematite, various amounts of SnCl$_2$*2H$_2$O were added to the aqueous electrolyte (12.5 mM FeCl$_3$, 50 mM KF, 0.1 M KCl, and 1 M H$_2$O$_2$). The molar SnCl$_2$ concentrations in the electrolyte were 5%, 10%, and 20% relative to the FeCl$_3$ concentration. Potentiostatic electrolysis was performed under conditions similar to those for the deposition of the pure hematite films. After drying, the samples were calcined, similar to treatment of $\alpha$-Fe$_2$O$_3$ films.

### 3.4. Characterization of the Samples: Study of the Phase Composition and Structure of Film Coatings

#### 3.4.1. X-ray Diffraction

The phase composition of the deposited oxide films was studied by X-ray diffraction (XRD) analysis using an Empyrean X-ray diffractometer (Panalytical BV, Malvern, UK). Ni-filtered Cu-K$\alpha$ radiation was used; the samples were studied in the Bragg–Brentano geometry. Experimental diffraction patterns were processed using the Highscore program; the phase composition was identified using the ICDD PDF-2 database.

#### 3.4.2. Absorption Spectra

Absorption spectra of the oxide films deposited on FTO glass slides were studied in the wavelength range of 300–700 nm at room temperature using a Lambda35 Perkin Elmer spectrometer (Waltham, MA, USA).

#### 3.4.3. Raman Spectra

Raman spectra were recorded using an inVia "Reflex" Raman spectrometer (Renishaw, New Mills Wotton-under-Edge, UK) with a 50× objective. Microphotographs were taken using a Leica microscope with a 50× objective (Wetzlar, Germany).

#### 3.4.4. SEM and EDX Measurement

The morphology of the deposited films and their microanalysis were assessed by scanning electron microscopy (SEM) with a JEOL JSM-6060 SEM and JED-2300 Analysis Station (JEOL, Tokyo, Japan).

### 3.4.5. Film Thickness Measurement

The thickness of the electrodeposited films was determined on a thin films measurement system MProbe 20 (Semiconsoft Inc., Southborough, MA, USA). The measurement range was from 1 nm to 1 mm.

### 3.4.6. AFM Measurement

The surface morphology was studied using a SolverPro scanning probe microscope (NT-MDT, Zelenograd, Russia) in the atomic force microscope (AFM) mode in air, the so-called ex situ AFM. For measurements, we used an NSG01 Golden silicon probe NT-MDT cantilever, size $125 \times 30 \times 2$ µm, with a resonant frequency in the range of 87–230 kHz and a force constant of 1.45–15.1 n/m. A square field was scanned at a rate of 0.1 Hz (0.1 lines per second) in the semi-contact mode.

### 3.4.7. XPS Measurement

The XPS studies were carried out on an OMICRON ESCA+ spectrometer (Taunusstein, Germany) with an aluminum anode equipped with a monochromatic X-ray source XM1000 (AlK$\alpha$ 1486.6 eV and power 252 W). To eliminate the local charge on the analyzed surface, a CN-10 charge neutralizer was used with an emission current of 2 µA and a beam energy of 1 eV. Argus was used as an analyzer–detector. The analyzer transmission energy was 20 eV, the scanning step on the binding energy scale was 0.1 eV, and the dwell time was 0.5 s. The spectrometer was calibrated using the Au4f 7/2 line at 84.1 eV. The pressure in the analyzer chamber did not exceed 10–9 mbar. All spectra were accumulated at least three times. The transmission energy was 20 eV. Background subtraction was performed using the Shirley method [62]. The iron states were adjusted according to [63].

### 3.4.8. Photoelectrochemical Measurements

A setup consisting of a PECC-2 photoelectrochemical three-electrode cell (Zahner Elektrik, Kronach, Germany), a solar spectrum simulator 96000 (Newport, Irvine, CA USA) equipped with an AM1.5G filter, with a power of 150 W, and an IPC-Pro MF potentiostat (IPCE RAS, Moscow, Russia) were used. The working electrode in the cell was FTO glass of 1 cm$^2$ area coated with hematite or Sn-doped hematite. The counter electrode was a platinum mesh with a geometric area of ~3 cm$^2$. An Ag/AgCl-KCl$_{std}$ reference electrode was used. The potential relative to the reversible hydrogen electrode was calculated using equation:

$$E_{RHE} = E_{Ag/AgCl} + 0.059 \, \text{pH} + E^o{}_{Ag/AgCl}, \text{ where } E^o{}_{Ag/AgCl} = 0.197 \text{ V}.$$

The photoanode was irradiated from the rear side. The light power was determined using a Nova instrument (OPHIR-SPIRICON LLC. Logan, UT, US). Photoelectrochemical oxidation of water and alcohols was carried out under visible light irradiation (1 Sun) with a 100 mW cm$^{-2}$ power density.

A computerized photoelectrochemical station, Zahner PP 211 CIMPS (Zahner-Elektrik Gmbh & Co.KG, Kronach, Germany), was used to measure the IMPS spectra of the photocurrent in the light modulation frequency range of 1000 to 0.02 Hz. The station was equipped with a TLS03 monochromatic light source with a set of LEDs with wavelengths in the range of 320–1020 nm and the CIMPS-QE/IPCE software package. In IMPS measurements, a photoanode was irradiated by monochromatic light with a wavelength λ of 452 nm. A sinusoidal disturbance (~10% of stationary light beam intensity) was superimposed on a stationary light with 14 mW cm$^{-2}$ power. Normalized IMPS curves were obtained by dividing the real and imaginary components of the experimental IMPS curve ($Re(I_{ph})$ and $Im(I_{ph})$) by the value of $I_2$. $I_2$ corresponds to the maximum value ($Re(I_{ph})$).

## 4. Conclusions

This study shows that the modification of the electrodeposited thin film hematite photoelectrodes with Sn enabled increasing the photoelectrochemical activity of alcohol oxidation by an order of magnitude. Intensity-modulated photocurrent spectroscopy reveals competitive paths of consumption of photoexcited holes—by recombination via surface states and by charge transfer through the photoanode/electrolyte interface. Hematite modification by Sn increased the rate of charge transfer through the photoanode/electrolyte interface to the loss of recombination rate.

Alcohols in 0.1 M KOH are found to be more effective acceptors of photoexcited holes in Sn-doped hematite compared with water molecules. The increase in the photoelectrooxidation rate in the sequence $H_2O < CH_3OH < C_2H_4(OH)_2 < C_3H_5(OH)_3$ was explained both by the increase in the rate of transfer of the photoexcited holes to the acceptors adsorbed at the photoanode surface and by the decrease in the recombination rate of the holes via surface states at the tin-doped hematite photoanode.

It is shown that the thin film Sn-modified hematite photoanodes are promising instruments for the photoelectrochemical degradation of organic pollutants.

**Supplementary Materials:** The following supporting information can be downloaded at: https://www.mdpi.com/article/10.3390/catal13111397/s1, Figure S1: Cyclic voltammograms on glassy carbon electrode ($0.0078$ cm$^{-2}$): (1)—25 mM SnCl$_2$–0.1 M KCl (1); (2)—25 mM SnCl$_2$–0.1 M KCl + 1 M H$_2$O$_2$. Scan rate; Figure S2: Cyclic voltammogram of SnCl$_2$ (7.5 mM) on glassy carbon electrode ($0.0078$ cm$^{-2}$) in 0.1 M KCl + 1 M H$_2$O$_2$. Scan rate 0.1 V s$^{-1}$.; Figure S.3: Cyclic voltammogram of SnCl$_2$ (25 mM) on glassy carbon electrode ($0.0078$ cm$^{-2}$) (1) and in the presence of 1M H$_2$O$_2$ (2) in 0.1M KCl. Scan rate 0.1 V s$^{-1}$.; Figure S4.a: SEM and EDX spectrum of ($\alpha$-Fe$_2$O$_3$ + 10% Sn$^{4+}$) film on a glassy carbon plate; Figure S4b: SEM and EDX spectrum of ($\alpha$-Fe$_2$O$_3$ + 10% Sn$^{4+}$) film, excluding carbon of substrate; Figure S5 XPS spectra of ($\alpha$-Fe$_2$O$_3$ + 10% Sn$^{4+}$) film on FTO. Spectrum of tin Sn3d (a), Fe2p3/2 and Sn3p3/2 regions (b), and oxygen O1s (c); Figure S6: Normalized absorption spectra for the film photoanodes: (1)—($\alpha$-Fe$_2$O$_3$); (2)—($\alpha$-Fe$_2$O$_3$ + 5% Sn$^{4+}$); (3)—($\alpha$-Fe$_2$O$_3$ + 10% Sn$^{4+}$); and (4)—($\alpha$-Fe$_2$O$_3$ + 20% Sn$^{4+}$); Figure S7: Voltammograms of the ($\alpha$-Fe$_2$O$_3$) film photoanode: (1)—dark voltammetry in 0.1 M KOH; (2–5)—voltammetry curves measured under visible light illumination with a power density of 100 mW cm$^{-2}$ in aqueous solutions. The measurements were performed in electrolytes: (2)—0.1 M KOH; (3)—0.1M KOH + 20% CH$_3$OH; (4)—0.1 M KOH + 20% C$_2$H$_4$(OH)$_2$; (5)—0.1 M KOH + 20% C$_3$H$_5$(OH)$_3$. The potential scan rate was 10 mV s$^{-1}$. Voltammograms under dark conditions for the alcohol-containing electrolytes solutions practically coincided with (1); Figure S8: Voltammograms of the ($\alpha$-Fe$_2$O$_3$ + 5% Sn$^{4+}$) film photoanode: (1)—dark voltammetry in 0.1 M KOH; (2–5) under visible light illumination with a power density of 100 mW cm$^{-2}$. The measurements were performed in electrolytes: (2)—0.1 M KOH; (3)—0.1 M KOH + 20% CH$_3$OH; (4)—0.1 M KOH + 20% C$_2$H$_4$(OH)$_2$; and (5)—0.1 M KOH + 20% C$_3$H$_5$(OH)$_3$. The potential scan rate was 10 mV s$^{-1}$. Voltammograms under dark conditions for the alcohol-containing electrolyte solutions practically coincided with (1); Figure S9: Voltammograms of the ($\alpha$-Fe$_2$O$_3$ + 20% Sn$^{4+}$) film photoanode: (1) dark voltammetry with 0.1 M KOH; (2–5)—under visible light illumination with a power density of 100 mW cm$^{-2}$. The measurements were performed in electrolytes: (2)—0.1 M KOH; (3)—0.1 M KOH + 20% CH$_3$OH; (4)—0.1 M KOH + 20% C$_2$H$_4$(OH)$_2$; and (5)—0.1 M KOH + 20% C$_3$H$_5$(OH)$_3$. The potential scan rate was 10 mV s$^{-1}$. Voltammograms under dark conditions for the alcohol-containing electrolytes solutions practically coincided with (1); Figure S10: Voltammograms of the ($\alpha$-Fe$_2$O$_3$ + 10% Sn$^{4+}$) 75 s film photoanode: (1) dark voltammetry 0.1 M KOH; (2–5)—under visible light illumination with a power density of 100 mW cm$^{-2}$. The measurements were performed in electrolytes: (2)—0.1 M KOH; (3)—0.1 M KOH + 20% CH$_3$OH; (4)—0.1 M KOH + 20% C$_2$H$_4$(OH)$_2$; and (5)—0.1 M KOH + 20% C$_3$H$_5$(OH)$_3$. The potential scan rate was 10 mV s$^{-1}$. Voltammograms under dark conditions for the alcohol-containing electrolyte solutions practically coincided with (1); Figure S11: Photocurrent variation with time measured with ($\alpha$-Fe$_2$O$_3$ + 10% Sn$^{4+}$) film photoanode at $E = 0.4$ V—(1) and at $E = 0.6$ V vs. Ag/AgCl—(2) under visible light illumination with a power density of 100 mW cm$^{-2}$ in 0.1 M KOH + 20% C$_3$H$_5$(OH)$_3$; Figure S12: IMPS curve measured for the ($\alpha$-Fe$_2$O$_3$ + 10% Sn$^{4+}$) photoanode illuminated with a monochromatic light of 453 nm at an illumination power density of 14 mW cm$^{-2}$ at $E = 0.5$ V vs. Ag/AgCl in 0.1 M KOH.

**Author Contributions:** V.A.G.: General conceptualization and preparation of film photoanodes. V.V.E.: conceptualization, preparation of photoanodes, and IMPS investigation: A.D.M.: photoelectrochemical measurements. A.A.A.: UV and Raman spectroscopic measurements. A.A.S.: XRD and X-ray fluorescence studies. I.G.B.: ASM measurement. A.V.S.: SEM and EDX measurement. All authors have read and agreed to the published version of the manuscript.

**Funding:** This research received no external funding.

**Data Availability Statement:** Not applicable.

**Acknowledgments:** This research was performed according to the IPCE RAS state assignment using the equipment of the Center for Collective Use of Physical Investigation Methods of the IPCE RAS.

**Conflicts of Interest:** The authors declare that they have no known competing financial interests or personal relationships that could have appeared to influence the work reported in this paper.

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
