# Peer review of "Sn-Doped Hematite Films as Photoanodes for Photoelectrochemical Alcohol Oxidation"

_catalysts, doi:10.3390/catal13111397_

Round 1

Reviewer 1 Report

Comments and Suggestions for Authors

Comments:

This article prepared Sn-doped hematite films for the photoelectrochemical oxidation of organic compounds such as methanol, ethylene glycol, and glycerol. The subject is interesting. However, there are many problems exist in the article, and more comprehensive discussion or explanations should be added. The work needs to be carefully checked in terms of sentence construction, grammar, and coherence of ideas. I think it needs a major revision before further consideration. Specific comments are as follows.

1.     Keywords.

The first one is confusing.

2.     Introduction

A large part of the introduction is devoted to describing the background of the foundation. The innovativeness of this work and the relevance of other work to this work is not reflected. The main concern is why alcohols are used as reactants. Alcohols are not harmful organics or pollutants in the traditional sense.

3.     Grammar and detail mistakes

There are many grammar and detail mistakes in the article, and this article needs to be carefully revised. It is strongly recommended that the authors invite native English-speaking scholars for linguistic checks. The following are typical examples:

(1) Page 1, line 32. There should be a comma after devices.

(2) Page 2, line 70. ‘28 mAcm-2and’. Note the spaces.

(3) Page 3, line 95. ‘tin’ or ‘Sn’. It's better to be consistent.

(4) Page 3, Line 99. ‘o C’. Inaccurate symbols for °C

4.     All Figures quality needs to be improved.

5.     The XRD graphs (Figure1) should be drawn in a more aesthetic and standardized way. For example, the positions of the different peaks and their corresponding crystal planes should be indicated.

6.     Page 4, line 127. Please confirm whether it is FeO or Fe3O4.

7.     Page 5, line 141-154. The presentation of results is logically confusing.

8.     Figure 3 (a). The use of (a.u.) for the vertical coordinate eliminates the need for a specific numerical scale.

9.     Page 7, line 166. How thickness is determined.

10.  Page 5, line 184-185. The substance of FTO is SnO2. Therefore, this statement is not valid.

11.  More direct evidence of Sn incorporation is needed, as well as the form in which Sn is present after incorporation

12.  Figure 11, please compare the IPCE under other conditions.

13.  The results should be expanded significantly and quantitatively.

14.  Authors shall carry out more studies to compare the results from this paper to that from other similar studies.

15.  Please give more details of the calculations and cite representative literature in the IMPS section.

Comments on the Quality of English Language

There are many grammar and detail mistakes in the article, and this article needs to be carefully revised. It is strongly recommended that the authors invite native English-speaking scholars for linguistic checks.

Author Response

Response to comments raised by Reviewer 1.

Recommendation Major Revision

This article prepared Sn-doped hematite films for the photoelectrochemical oxidation of organic compounds such as methanol, ethylene glycol, and glycerol. The subject is interesting. However, there are many problems exist in the article, and more comprehensive discussion or explanations should be added. The work needs to be carefully checked in terms of sentence construction, grammar, and coherence of ideas. I think it needs a major revision before further consideration.

We are very grateful to the Reviewer for through reading of the manuscript and valuable comments. The comments undoubtedly helped us improve quality of the manuscript.

Specific comments are as follows.

  1.  

The first is confusing.

Response. Corrected.

Introduction

A large part of the introduction is devoted to describing the background of the foundation. The innovativeness of this work and the relevance of other work to this work is not reflected. The main concern is why alcohols are used as reactants. Alcohols are not harmful organics or pollutants in the traditional sense.

Response. Relevant clarifications have been added to the introduction.

  1. Grammar and detail mistakes

There are many grammar and detail mistakes in the article, and this article needs to be carefully revised. It is strongly recommended that the authors invite native English-speaking scholars for linguistic checks. The following are typical examples:

  • Page 1, line 32. There should be a comma after devices.

Response. The text has been corrected and a comma has been added.

  • Page 2, line 70. ‘28 mAcm-2and’. Note the spaces.

Response. Spaces were inserted between words throughout the article.

(3) Page 3, line 95. ‘tin’ or ‘Sn’. It's better to be consistent.

Response. Corrected.

  • Page 3, Line 99. ‘o C’. Inaccurate symbols for °C.

Response. Corrected here and on page 15.

English language checked and corrected.

  1. All Figures quality needs to be improved.

Response. All figures of high quality are presented in additional file in format tiff.

  1. The XRD graphs (Figure1) should be drawn in a more aesthetic and standardized way. For example, the positions of the different peaks and their corresponding crystal planes 2 should be indicated.

Response. The figure with X-ray data is redrawn.

  1. Page 4, line 127. Please confirm whether it is FeO or Fe3O4.

Response. We confirm Fe3O4.

  1. Page 5, line 141-154. The presentation of results is logically confusing.

The band gap of the electrodeposited films was estimated using Tauc coordinates [58,59] constructed for normalized curves (see Fig. 3b).

Response. Corrected.

  1. Figure 3 (a). The use of (a.u.) for the vertical coordinate eliminates the need for a specific numerical scale.

Response. We agree with the Reviewer. The marks are not removed, although they are meaningless.

  1. Page 7, line 166. How thickness is determined.

Response. The thickness of the electrodeposited films was determined on a Thin Films measurement system MProbe 20 (Semiconsoft Inc. USA). Film thickness decreases with the increase of Sn+4 concentration in the electrolyte. Thicknesses of (α-Fe2O3), (α-Fe2O3 +5% Sn4+), (α-Fe2O3 +10%Sn4+), and (α-Fe2O3 +20% Sn4+) films  are roughly ~300 nm, ~250, ~200, and ~150 nm, respectively.

Explanations are given in p.8

2.4. SEM images  

Instead of 2.4. SEM images write 2.4. Morphology and thickness of the electrodeposited films

Response. Corrected

  1. Page 5, line 184-185. The substance of FTO is SnO2. Therefore, this statement is not valid.

Response. Corrected in p.4

  1. More direct evidence of Sn incorporation is needed, as well as the form in which Sn is present after incorporation.

Response. Additional measurements were performed. Results are given in Supplementary Materials pp.3-6, Figs.S4, S5.To directly prove the incorporation of Sn into the film and to exclude the influence of the conducting SnO2 substrate, a sample (α-Fe2O3+10% Sn4+) was prepared on a glassy carbon plate. The results of EDX analysis clearly confirm the presence of Sn in the film and are presented in Supplementary Materials. The film on the glassy carbon plate is prepared as described in paragraph 3.2 and 3.3. Based on the EDX and XPS analysis results, it can be assumed that tin is present as a solid solution in hematite.

Figure 11, please compare the IPCE under other conditions.

Response. We have measured IPCE at different electrode potentials and observed no difference. The IPCE curve was needed to estimate the wavelength in the visible light region providing high photocurrents for the IMPS study of recombination loss on a tin-modified hematite photoanode. In this case, the photoanode potential was the same for the IPCE and IMPS curves and corresponded to 0.5 V vs Ag/AgCl. At lower potentials, the accuracy of the IMPS curve measurements decreased.

  1. The results should be expanded significantly and quantitatively.

Response. We significantly increased Supplementary Materials section in efforts to provide more evidence. The results of proving the doping of hematite with tin were significantly supplemented by quantitative data.

  1. Authors shall carry out more studies to compare the results from this paper to that from other similar studies.

Response. We failed to find published data on the photoelectrooxidation of one, two and three atomic alcohols on hematite thin films with different doping concentration of Sn4+. Comparison with similar data that was measured by our group on films of hematite modified with Bi, Co, Ti and ZnO [26-28] shows that hematite doped with tin has the greatest photoelectrocatalytic activity in the photoelectrooxidation reaction of the studied alcohols. A comparison with literature data is given at the end of section 2.4.

  1. Please give more details of the calculations and cite representative literature in the IMPS section.

Response. References are added.

  1. Peter, L.M.; Ponomarev, E.A.; Fermin, D.J. Intensity-modulated photocurrent spectroscopy: Reconciliation of phenomenological analysis with multistep electron transfer mechanisms. J. Electroanal. Chem. 1997, 427, 79–96.
  2. Peter, L.M.;Wijayantha, K.G.U.; Tahir, A.A. Kinetics of light-driven oxygen evolution at a-Fe2O3 electrodes. J. Faraday Discuss. 2012, 155, 309–322.
  3. Klotz, D.; Ellis, D.S.; Dotana, H.; Rothschild, A. Empirical in operando Analysis of the Charge Carrier Dynamics in Hematite Photoanodes by PEIS, IMPS and IMVS. J. Phys. Chem. Chem. Phys. 2016, 18, 23438–23457.

Reviewer 2 Report

Comments and Suggestions for Authors

This manuscript report the preparation of hematite thin films with different doping concentration of Sn4+ and the corresponding PEC performance in photooxidtion of alcohols. This is an interesting research and should be of interest to the readers. I suggest major revision of the manuscript due to the following questions.

Line91-92,“Photoanodes with films of α- 91 Fe2O3 modified Sn4+were obtained at the same potential E=-0.35 V vs. Ag/AgCl (deposition 92 time 150 s) in the presence in the electrodeposition electrolyte of various amounts of SnCl4should be revised.

Figure 1, The XRD pattern of the Sn:Fe2O3 is not properly marked. The square should but put on the top of all of the peaks for hematite.

Line 118, “A separate SnO2 phase was not detected in the diffraction spectrum.” This sentence is confusing because the FTO substrate should have SnO2 diffraction.

Line 165, “The SEM image of (α-Fe2O3) film (1)” should be revised into “The SEM image of (α-Fe2O3) film (Fig. 4.1)”.

Line 165, ”With an increase in the concentration of tin in the electrolyte to 20% (4), the thickness of the hematite film decreases,” How can you conclude the change thickness from the SEM images? The same question also arises from the statement in line 202.

In Figure 4, there are big white “particles” in the graphs. What are they and where they are from?

In Figure 6 and 7, the applied bias is vs. Ag/AgCl. While the discussion in the main text is vs. RHE.

In general, the IPCE curves should be consisted of discontinuous dots, which is different from Fig.11.

The main contribution of this manuscript is the observation of trend of photodegradation of the alcohols. The Partial voltammetry curves were derived from the photocurrent difference between the presence and absence of alcohols. This process confusing because the photocurrent of alcohol electrolyte probably are due to the oxidation of purely alcohol instead of alcohol+water. The authors should explain this calculation method.

There seems to be a trend in the oxidation of the alcohols in PEC process. It is suggest to briefly discuss the relationship between the structure of the alcohols and the oxidation trend.

Comments on the Quality of English Language

The Englis is acceptable with minor polishing.

Author Response

Response to comments raised by Reviewer 2.

Recommendation Major Revision

We are very grateful to the reviewer who carefully read the manuscript and made valuable comments, the consideration of which will undoubtedly improve the quality of the manuscript.

This manuscript report the preparation of hematite thin films with different doping concentration of Sn4+ and the corresponding PEC performance in photooxidation of alcohols. This is an interesting research and should be of interest to the readers. I suggest major revision of the manuscript due to the following questions.

Line 91-92,“Photoanodes with films of α- Fe2O3 modified Sn4+ were obtained at the same potential E=-0.35 V vs. Ag/AgCl (deposition  time150 s) in the presence in the electrodeposition electrolyte of various amounts of SnCl4“ should be revised.”

Response. The deposition of the hematite film and the tin-modified hematite film was carried out at the same potential -0.35 V vs. Ag/AgCl.

Corrected has been made to the text.

Figure 1, The XRD pattern of the Sn:Fe2O3 is not properly marked. The square should but put on the top of all of the peaks for hematite.

Response. Corrected in the text.

Line 118, “A separate SnO2 phase was not detected in the diffraction spectrum.” This sentence is confusing because the FTO substrate should have SnO2 diffraction.

Response. Corrected in the text.

Line 165, “The SEM image of (α-Fe2O3) film (1)” should be revised into“The SEM image of (α-Fe2O3) film (Fig. 4.1)”.

Response. Corrected in the text.

Line 165, ”With an increase in the concentration of tin in the electrolyte to 20% (4), the thickness of the hematite film decreases,” How can you conclude the change thickness from the SEM images? The same question also arises from the statement in line 202.

Response. The thickness of the electrodeposited films was determined on a Thin Films measurement system MProbe 20 (Semiconsoft Inc. USA). Film thickness decreases with the increase of Sn+4 concentration in the electrolyte. Thicknesses of (α-Fe2O3), (α-Fe2O3 +5% Sn4+), (α-Fe2O3 +10%Sn4+), and (α-Fe2O3 +20% Sn4+) films are roughly ~300 nm, ~250, ~200, and ~150 nm, respectively.

Corrected in the text.

In Figure 4, there are big white “particles” in the graphs. What are they and where they are from?

Response. White blebs are clusters of Fe-oxide nanoparticles. Their exact composition is difficult to determine due to small size. High brightness in the SE mode is explained by pronounced topography and well-developed surface providing numerous spots with enhanced emission of secondary electrons. The same blebs are also visible on AFM scans, see Fig. 5 (3,4)

In Figure 6 and 7, the applied bias is vs. Ag/AgCl. While the discussion in the main text is vs. RHE.

Response. Page 9, line 213, instead at E=1.23 V (vs. RHE) need to write

E=0.5 V (vs. Ag/AgCl)

Corrected in the text.

Page 9, line 225, 231 instead at E=1.23 V (vs. RHE) need to write

at E=1.23 V vs. RHE (which correspond to E=0.27 V vs. Ag/AgCl)

Corrected in the text.

In general, the IPCE curves should be consisted of discontinuous dots, which is different from Fig.11.

Response. Fig. 11 is corrected

The main contribution of this manuscript is the observation of trend of photodegradation of the alcohols. The Partial voltammetry curves were derived from the photocurrent difference between the presence and absence of alcohols. This process confusing because the photocurrent of alcohol electrolyte probably are due to the oxidation of purely alcohol instead of alcohol+water. The authors should explain this calculation method.

Response. In alcohol solutions, photoelectrooxidation of water and alcohols are competing processes. Moreover, a smaller part of the total photocurrent is due to the oxidation of water, and the majority of the photocurrent is associated with the photoelectrooxidation of alcohols. The contribution of the photocurrent of photoelectrooxidation of water in alcohol solutions (Fig. 8-9) due to competition is even lower than in 0.1 M KOH. In view of the small contribution of the process of photoelectrooxidation of water to the total photocurrent (Fig.7), the partial curves obtained from the difference in photocurrents in the presence and absence of alcohols, to a first approximation, correctly reflect the photocurrents of photoelectrooxidation of alcohols.

There seems to be a trend in the oxidation of the alcohols in PEC process. It is suggest to briefly discuss the relationship between the structure of the alcohols and the oxidation trend.

Response. On doped tin hematite, upon transition from one-, two-, and three-atomic alcohols, an increase in the photocurrent of photoelectrooxidation is observed. This can be explained by an increase in the number of adsorption sites for an individual alcohol molecule, which facilitates the transfer of a hole to the depolarizer.

Round 2

Reviewer 1 Report

Comments and Suggestions for Authors

All concerns have been addressed and are acceptable for publication.

Author Response

We thank the Reveiwer for thorough work with our manuscript and creative criticism.  We believe that amendments substantially improved quality and readability of our manuscript.

Reviewer 2 Report

Comments and Suggestions for Authors I think all the revsion suggestion has been addressed. I suggest acceptance of the manuscript after minor revision. In the answers of the author to this question: In Figure 4, there are big white “particles” in the graphs. What are they and where they are from? Response. White blebs are clusters of Fe-oxide nanoparticles. Their exact composition is difficult to determine due to small size. High brightness in the SE mode is explained by pronounced topography and well-developed surface providing numerous spots with enhanced emission of secondary electrons. The same blebs are also visible on AFM scans, see Fig. 5 (3,4) I suggest explanation of the white blebs (FIgure 4) in the manuscrit because this morphology is obvious, which derserve an explanation in the text. Comments on the Quality of English Language

No comment on the English. Good enough.

Author Response

We thank the Reviewer 2 for thorough work on our manuscript and creative criticism. We believe that amendments improved quality and readability of the manuscript.

Remark.

I think all the revision suggestion has been addressed. I suggest acceptance of the manuscript after minor revision. In the answers of the author to this question: In Figure 4, there are big white “particles” in the graphs. What are they and where they are from? Response. White blebs are clusters of Fe-oxide nanoparticles. Their exact composition is difficult to determine due to small size. High brightness in the SE mode is explained by pronounced topography and well-developed surface providing numerous spots with enhanced emission of secondary electrons. The same blebs are also visible on AFM scans, see Fig. 5 (3,4) I suggest explanation of the white blebs (Figure 4) in the manuscript because this morphology is obvious, which deserve an explanation in the text.

Response.

We revised the text. The following explanation is placed in lines 178-183. It is highlighted.

Big white “particles” in Fig. 4. are clusters of Fe-oxide nanoparticles. Their exact composition is difficult to determine because of rather small size. High brightness in the SE mode is due to pronounced topography and well-developed surface providing numerous spots with enhanced emission of secondary electrons. The same “particles” are also visible on AFM scans, see Fig. 5).
